# Redox-mediated decoupled seawater direct splitting for $H_2$ production

Tao Liu [1,2,3,4,8] ✉, Cheng Lan [1,2,4,8] ✉, Min Tang[5,8], Mengxin Li[2,3], Yitao Xu[5], Hangrui Yang[6], Qingyue Deng[5], Wenchuan Jiang [1,2,4], Zhiyu Zhao[1,2,4], Yifan Wu [1,2,4] ✉ & Heping Xie [1,2,3,4,7] ✉

Seawater direct electrolysis (SDE) using renewable energy provides a sustainable pathway to harness abundant oceanic hydrogen resources. However, the side-reaction of the chlorine electro-oxidation reaction (ClOR) severely decreased direct electrolysis efficiency of seawater and gradually corrodes the anode. In this study, a redox-mediated strategy is introduced to suppress the ClOR, and a decoupled seawater direct electrolysis (DSDE) system incorporating a separate $O_2$ evolution reactor is established. Ferricyanide/ferrocyanide ($[Fe(CN)_6]^{3-/4-}$) serves as an electron-mediator between the cell and the reactor, thereby enabling a more dynamically favorable half-reaction to supplant the traditional oxygen evolution reaction (OER). This alteration involves a straightforward, single-electron-transfer anodic reaction without gas precipitation and effectively eliminates the generation of chlorine-containing byproducts. By operating at low voltages (~1.37 V at 10 mA cm$^{-2}$ and ~1.57 V at 100 mA cm$^{-2}$) and maintaining stability even in a Cl$^-$-saturated seawater electrolyte, this system has the potential of undergoing decoupled seawater electrolysis with zero chlorine emissions. Further improvements in the high-performance redox-mediators and catalysts can provide enhanced cost-effectiveness and sustainability of the DSDE system.

Hydrogen is widely considered a promising clean energy source, with a superior energy density of 142.351 MJ kg$^{-1}$ and the ability to burn without emitting greenhouse gases[1,2]. Electrolytic seawater splitting provides a potentially compelling way to convert renewable energy sources in the ocean, such as wind and solar power, into hydrogen energy, thus producing a viable energy-saving alternative while addressing the challenge of renewable energy instability[3–6].

To date, hydrogen production directly from seawater resources remains to be further explored, and the electrochemical oxidation reaction of the chloride ions (Cl$^-$) on the anode poses a preeminent challenge to the advancement of seawater direct electrolysis technology[5,7]. Specifically, the occurrence of the chlorine electro-oxidation reaction (ClOR), which competes with the oxygen evolution reaction (OER), prompts the emergence of corrosive ions and noxious

[1]State Key Laboratory of Intelligent Construction and Healthy Operation and Maintenance of Deep Underground Engineering, Sichuan University & Shenzhen University, Chengdu 610065, China. [2]Institute of New Energy and Low-Carbon Technology, Sichuan University, Chengdu 610065, China. [3]Guangdong Provincial Key Laboratory of Deep Earth Sciences and Geothermal Energy Exploitation and Utilization, Institute of Deep Earth Sciences and Green Energy, Shenzhen University, Shenzhen 518060, China. [4]Shenzhen Key Laboratory of Deep Engineering Science and Green Energy, Institute of Deep Earth Sciences and Green Energy, Shenzhen University, Shenzhen 518060, China. [5]Sichuan University-Pittsburgh Institute, Chengdu 610065, China. [6]School of Chemical Engineering, Sichuan University, Chengdu 610065, China. [7]College of Water Resource & Hydropower, Sichuan University, Chengdu 610065, China. [8]These authors contributed equally: Tao Liu, Cheng Lan, Min Tang. ✉e-mail: liutao3200023@scu.edu.cn; lancheng@scu.edu.cn; fairwu@qq.com; xiehp@scu.edu.cn

gases (ClO⁻, Cl₂, HOCl, etc.). This poses a severe threat to the ecosystem while simultaneously leading to electrode corrosion and decreasing the efficiency and sustainability of the electrolysis process[8,9]. Considering the reaction kinetics, the OER has a potential advantage (~490 mV) over the ClOR under alkaline conditions because of its standard redox potential[6,10]. Therefore, corrosion-resistant alkaline electrocatalysts have been developed to enable the core electrolytic assembly to withstand the complex composition of the seawater direct electrolytic system; these electrocatalysts included transition metal oxides[11,12], nitrides[9,13], phosphates[14,15], and hydroxides[16–18]. Unfortunately, continuous seawater injection into electrolyzers results in Cl⁻ accumulation to saturation, gradually corroding the anode and the entire system during electrolysis[19,20].

An alternative approach to SDE involves replacing the OER with a more thermodynamically favorable electro-oxidation reaction to overcome the impact of ClOR[21]. Qiu et al. proposed a seawater splitting method by coupling the HER with novel anodic reactions such as hydrazine degradation[22] and the sulfion oxidation reaction[23]. Species more susceptible to oxidation, such as $N_2H_4$ and $S^{2-}$, have been introduced as reducing agents to replace the OER, reducing the voltage of the electrolysis system and effectively preventing the consequences of the chlorine electro-oxidation reaction thermodynamically[24,25]. However, this strategy inevitably involves the consumption of raw chemicals and the purification of oxidation products; therefore, additional costs to the entire electrolysis system are incurred[26].

Decoupled water electrolysis is a novel approach that promotes stepwise reactions by using an auxiliary redox mediator serving as an electron/ion buffer, thereby separating the HER and OER in space and time[27,28]. This decoupled strategy is typically used to address limitations such as low load, high voltage, and gas purity in electrolytic cells[29,30]. In this study, we discovered that the stepwise reactions of the decoupled strategy demonstrated tremendous potential for designing the oxidation reaction of the redox mediator at the anode. Moreover, the hysteretic four-electron kinetics of the OER was the critical factor responsible for generating a large overpotential, severely limiting the energy conversion efficiency and stability of the seawater electrolyzer[1,31]. Therefore, another alternative strategy to replace the electrochemical OER against the ClOR could be to design a more dynamically favorable electro-oxidation reaction at the anode; this favorable reaction could be achieved by this decoupled approach.

Here, we propose a specific decoupled strategy for seawater splitting and design a decoupled seawater direct electrolysis (DSDE) system for the hydrogen production directly from seawater; this system includes a redox-flow seawater direct electrolysis cell integrated with a separate O₂ evolution reactor (Fig. 1). This system can effectively prevent the occurrence of the ClOR and avoid the generation of chlorine-containing substances without the consumption of additional chemicals. In the system (Fig. 1a), the electrolyte-borne redox mediator ferricyanide/ferrocyanide ([Fe(CN)₆]³⁻/⁴⁻) dissolved in alkaline seawater and circulates between the electrochemical cell and the individual reactor as charge carriers. In the electrochemical cell (Fig. 1b), the HER (Eq. 1) is maintained at the cathode, and the ferricyanide electro-oxidation reaction (Eq. 2) occurs at the anode, which has both thermodynamic and kinetic advantages over the ClOR. Meanwhile, in the individual chemical reduction reactor (Fig. 1c), the electron-losing [Fe(CN)₆]³⁻ and OH⁻ spontaneously undergo a reduction reaction (Eq. 3) and release O₂ facilitated by the Fe-Ni(OH)₂/NF catalyst (Fig. 1d). During this process, an anion exchange membrane is used to separate the anode and the cathode, and to ensure the overall electrical neutrality of the system, OH⁻ migrates from the cathode to the anode through the anion exchange membrane (AEM). The experimental results indicate that the [Fe(CN)₆]³⁻/⁴⁻ redox couple maintains reversible redox kinetics and stability in seawater. The DSDE system operates at low voltages of approximately -1.37 V at 10 mA cm⁻² and -1.57 V at 100 mA cm⁻² and demonstrates stability at a near-industrial current

density of 200 mA cm⁻² for more than 250 h. In addition, the system also effectively performs in Cl⁻-saturated alkaline seawater. On this basis, decoupled seawater electrolysis can be further achieved by designing redox mediators with high capacity and suitable potential for better cost-effectiveness and sustainability.

$$Cathode : 2H_2O + 2e^- \rightarrow H_2 + 2OH^- \tag{1}$$

$$Anode : 2\left[Fe(CN)_6\right]^{4-} - 2e^- \rightarrow 2\left[Fe(CN)_6\right]^{3-} \tag{2}$$

$$Chemical\ Reduction\ Reactor :$$
$$4\left[Fe(CN)_6\right]^{3-} + 4OH^- \rightarrow 4\left[Fe(CN)_6\right]^{4-} \tag{3}$$
$$+ 2H_2O + O_2$$

## Results and discussion
### Verification of the redox characteristics in seawater

[Fe(CN)₆]³⁻/⁴⁻ (Fig. 2a) is well known for its robustness as a redox mediator, with abundant, stable, and nontoxic characteristics and favorable electrochemical features[32,33]. Therefore, it has extensive applications in fundamental electrochemical studies, high-performance redox-flow batteries, and other fields[32–35]. In the decoupled seawater electrolysis system, the process of water electrolysis is integrated with an additional catalytic chemical reaction and needs redox mediators that are characterized by exceptional redox cyclability, strong stability in alkaline seawater, and a theoretical redox potential exceeding the OER potential (1.23 V vs. RHE) with the actual potential located below the oxygen production on the electrode. Notably, [Fe(CN)₆]³⁻/⁴⁻ precisely meets these criteria; thus, it is a suitable redox mediator for the complex scenario of decoupled real seawater electrolysis. The cyclic voltammogram (CV) of the redox mediator [Fe(CN)₆]³⁻/⁴⁻ is expected to occur between the linear sweep voltammetry (LSV) curves of the HER and the OER, aligning with the fundamental principle of achieving electrochemical hydrogen production and regeneration of the redox carrier. Hence, CV curve experiments were conducted on 10 mM [Fe(CN)₆]³⁻ solutions in three different electrolytes: alkaline freshwater (1 M KOH, pH = 13.59 ± 0.13), alkaline simulated seawater (1 M KOH + 0.5 M NaCl, pH = 13.54 ± 0.14), and alkaline seawater (1 M KOH + seawater, seawater comes from Shenzhen Bay, China, pH = 13.49 ± 0.19, the composition in Table S1), with linear LSV measurements performed without [Fe(CN)₆]³⁻ in these solutions (Fig. 2a). Across all three electrolytes, [Fe(CN)₆]³⁻/⁴⁻ showed a pair of reversible redox peaks within the potential range of 0.9 to 1.6 V, and these peaks occurred at a potential of -1.27 V (vs. RHE). Notably, this redox process remained within the true potential of the HER and OER, effectively avoiding the chlorine oxidation reaction. Furthermore, CV experiments of [Fe(CN)₆]³⁻/⁴⁻ at various scan rates were performed (Fig. 2c and Fig. S1). Notably, the peak current of [Fe(CN)₆]³⁻/⁴⁻ in seawater exhibited a linear relationship with half of the scan rate, and the ratio of the peak oxidation current to the peak reduction current $i_{pa}:i_{pc}$ was -1 (Fig. 2d); these results further indicated that [Fe(CN)₆]³⁻/⁴⁻ exhibited quasi-reversible redox characteristics in seawater.

The electrochemical redox stability of [Fe(CN)₆]³⁻/⁴⁻ in the aforementioned electrolytes was subsequently verified through the cyclic scanning experiments (Fig. 2e and Fig. S2). Remarkably, even after 5000 cyclic voltammetry scans, the peak pattern remained consistent, with the attenuation of this redox reaction remaining below 5% in seawater, demonstrating its robust and enduring electrochemical reversibility. Furthermore, we conducted a rotating disk electrode experiment to assess the electrochemical oxidation kinetics of ferricyanide in seawater (Fig. 2f). By selecting the current corresponding to 0.5 V (vs. Ag/AgCl) at various rates as the limiting current ($i_{lim}$), we

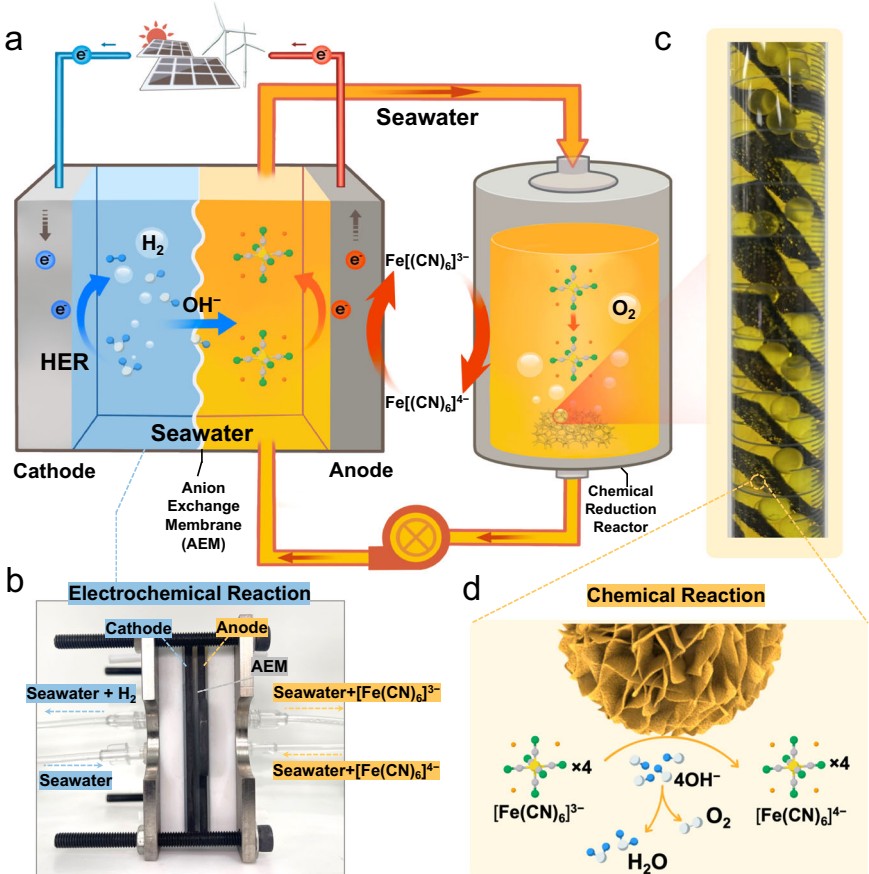

**Fig. 1 | Decoupled seawater direct electrolysis strategy. a** Schematic diagram of the redox-flow decoupled seawater direct electrolysis strategy. The blue, white, gray, yellow, green, and orange balls represent the H atoms, O atoms, C atoms, Fe atoms, N atoms, and electrons, respectively. **b** Photograph of the electrochemical cell, with HER at the cathode and $[Fe(CN)_6]^{4-}$ being converted to $[Fe(CN)_6]^{3-}$ at the anode. **c** Photograph of the $O_2$ gas evolution in the separate reactor. **d** Schematic of the reaction process at the catalyst interface. $[Fe(CN)_6]^{3-}$ and $OH^-$ spontaneously undergo a reduction reaction on the Fe-Ni(OH)$_2$/NF catalyst.

observed a strong positive correlation between $i_{lim}$ and the square root of the rotation rates: $y = 0.00694x$, $R^2 = 0.999$; these results indicated diffusion-controlled oxidation (Fig. 2g). The diffusion coefficient of $[Fe(CN)_6]^{4-}$ oxidation in the seawater electrolyte was calculated by using the Koutecky–Levich equation: $D_O = 4.496 \times 10^{-6}$ cm$^2$ s$^{-1}$. The Koutecky–Levich plots confirmed a good linear correlation between the reciprocal of the kinetically limited current and the inverse square root of the rotation rates (Fig. 2h). The exchange current ($i_O$) was calculated to be 0.201 mA cm$^{-2}$ by fitting the Tafel plot with the Butler–Volmer equation (Fig. 2i).

### Characterization of the electrochemical reaction

The electrochemical redox characteristics of the mediator in seawater have been extensively investigated. We further explored the electrochemical process of the DSDE system. The oxidation process of ferrocyanide at the practical electrode in the H-type electrolytic cell was initially investigated. In the experiment, a standard configuration was used; here, the graphite rod served as the counter electrode and the Hg/HgO electrode was used as the reference electrode. Importantly, the practical working electrode was a pretreated graphite felt that benefited from the ability of the redox carrier to swiftly capture electrons. Since this process utilized cost-effective carbon materials such as carbon felt, the need for a metal electrocatalyst at the anode could be eliminated. Moreover, to ensure a sufficient supply of $OH^-$ for the chemical oxygen evolution process, three Ar-saturated electrolytes with 4 M KOH were chosen for subsequent experiments. As depicted in Fig. 3a, the oxidation LSV curves of 0.3 M $[Fe(CN)_6]^{4-}$ on the carbon felt electrode at 25 °C with a stirring rate of

500 r min$^{-1}$ were evaluated, and the LSV curves of the carbon felt electrode without $[Fe(CN)_6]^{4-}$ were obtained for the same three solutions for comparison. The oxidation potentials of $[Fe(CN)_6]^{4-}$ at the carbon felt electrode were 1.330 V in alkaline seawater (pH = 14.06 ± 0.21), 1.361 V in alkaline simulated seawater (pH = 14.12 ± 0.17), and 1.369 V in alkaline freshwater (pH = 14.21 ± 0.11) at a current density of 10 mA cm$^{-2}$. At current densities of up to 100 mA cm$^{-2}$, the carbon felt electrode required potentials of 1.428 V in alkaline seawater, 1.436 V in alkaline simulated seawater, and 1.457 V in alkaline freshwater; here, no oxygen evolution reaction occurs. The results demonstrated that the electrochemical oxidation rate of $[Fe(CN)_6]^{4-}$ in 4 M KOH + seawater was almost equal to that in 4 M KOH + 0.5 M NaCl and better than that in 4 M KOH. The viability of the decoupled seawater electrolysis strategy for producing hydrogen was further reinforced by the oxygen evolution potential of 2.165 V at identical current densities of 100 mA cm$^{-2}$ and showed a potential difference of no less than 0.708 V. Besides, the LSV curves without $iR$ compensation were shown in Fig. S3. Additionally, we explored the influence of temperature on the electrochemical oxidation reaction of $[Fe(CN)_6]^{4-}$. From Fig. S4, the oxidation reaction rate of the three electrolytes increased as the temperature increased. In seawater, at a current density of 100 mA cm$^{-2}$, the oxidation potential of $[Fe(CN)_6]^{4-}$ decreased from 1.428 V at 25 °C to 1.412 V at 70 °C. Similarly, the temperature increased the reaction rate of the HER[6]. Therefore, temperature had a positive effect on the electrochemical reaction rate of the decoupled seawater direct electrolysis system.

The stability of the $[Fe(CN)_6]^{4-}$ oxidation reaction was assessed by long-term electrolysis at various current densities in different

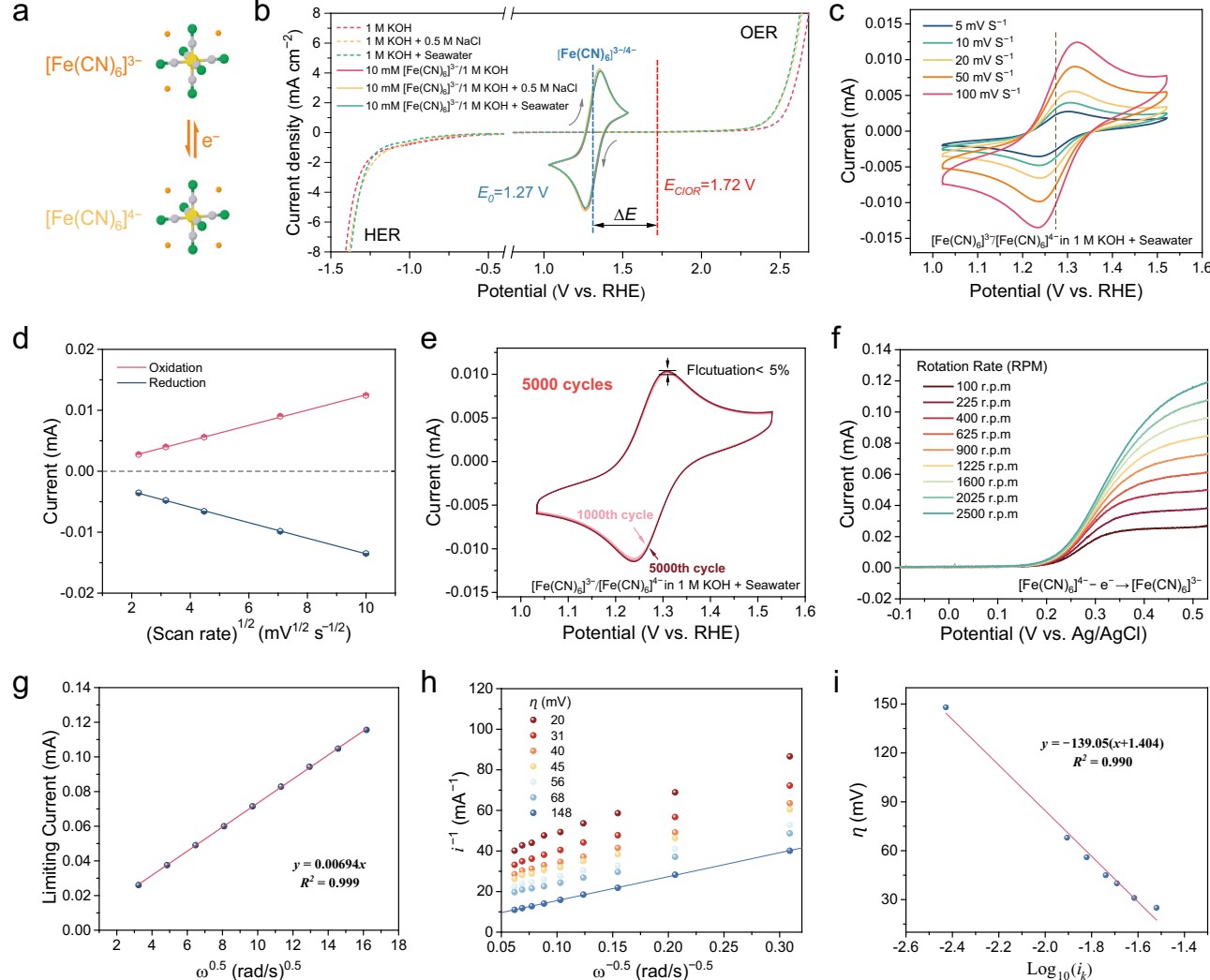

**Fig. 2 | Redox characterization of the $[Fe(CN)_6]^{3-/4-}$ mediator in seawater.** The experiments were carried out at room temperature (-25 °C). The carbon rod electrode serving as the counter electrode, and the Ag/AgCl electrode (in saturated KCl solution) acting as the reference electrode. **a** Redox reaction of $[Fe(CN)_6]^{3-/4-}$, gray, yellow, green, and orange balls represent the C atoms, Fe atoms, N atoms, and electrons, respectively. **b** CV curves of $[Fe(CN)_6]^{3-/4-}$ in 1 M KOH, 1 M KOH + 0.5 M NaCl, and 1 M KOH + seawater at 5 mV s⁻¹ and LSV curves without $[Fe(CN)_6]^{3-}$ in the above electrolytes without $iR$ compensation. The working electrode is a glassy carbon electrode. **c** CV curves at different scan rates in 1 M KOH + seawater and **d** the relationship between peak currents in 1 M KOH + seawater and $\omega^{1/2}$. **e** CV curves of $[Fe(CN)_6]^{3-/4-}$ in 1 M KOH + seawater over 5000 cycles at 50 mV s⁻¹. **f** RDE experiment of $[Fe(CN)_6]^{3-/4-}$ in 1 M KOH + seawater from 100 to 2500 rpm. **g** Levich plot of the measured limiting current versus the square root of the rotation rate ($\omega^{0.5}$). **h** Koutecky−Levich plot at different overpotentials from 20 to 148 mV. **i** Fitting the Tafel plot with the Butler–Volmer equation; here, the logarithm of the kinetically limited current is plotted against the overpotential.

solutions (Fig. 3b). The $[Fe(CN)_6]^{4-}$ oxidation reaction showed oxidation stability for more than 16 h. The reaction potentials of $[Fe(CN)_6]^{4-}$ in alkaline seawater and alkaline simulated seawater were slightly lower than that in alkaline freshwater. To investigate the impact of $Cl^-$ on the system, the stabilities of the carbon felt electrode at the anode at 50 mA cm⁻² in three different solutions with/without $[Fe(CN)_6]^{4-}$ were analyzed (Fig. 3c). In alkaline seawater, the potential continuously increased, and the carbon felt electrode fractured in less than 6 h of reaction due to electrochemical corrosion (Fig. S5). In $[Fe(CN)_6]^{4-}$ + alkaline seawater, the potential was more stable, and the full-range average voltage was -1.45 V. In addition, $ClO^-$ was not produced in the $[Fe(CN)_6]^{4-}$ + alkaline seawater under prolonged testing, thus eliminating anodic corrosion regardless of $Cl^-$ crossing (Fig. 3d and Fig. S6). Considering that $Cl^-$ continuously accumulated during the reaction, the LSV and V-T curves under $Cl^-$-saturated conditions (-3.16 M by ion chromatography) were further examined. In Fig. 3e, the oxidation potential of $[Fe(CN)_6]^{4-}$ slightly increased from 1.428 V to 1.439 V when $Cl^-$ was saturated at 100 mA cm⁻²; this value

was significantly lower than the theoretical potential of the ClOR. The original LSV curves with/without $iR$ compensation were shown in Fig. S7. The long-term stability test was run for 120 h (Fig. 3f), and the potential only slightly increased, with an overall average potential of 1.50 V. Throughout the entire electrolysis process, $ClO^-$ detection was conducted using an iodometric method-based instrument (Fig. S8), and the results revealed that $ClO^-$ did not form during the electrolysis process in $[Fe(CN)_6]^{4-}$ + $Cl^-$-saturated alkaline seawater. These findings showed the electro-oxidation efficiency and stability properties of $[Fe(CN)_6]^{4-}$ in carbon felt electrodes.

For the HER at the cathode, we selected our previously reported high-performance electrocatalyst, $(NiMo)_{1-x}Co_xP/NF$[36]. Its catalytic activity and long-term stability were evaluated in alkaline seawater at room temperature. Based on the results in Fig. S9a, b, a slight decrease in catalytic HER activity in alkaline seawater was observed, and the overpotential attained a current density of 100 mA cm⁻² at 197 mV; this value is slightly higher than that in alkaline freshwater (176 mV). A galvanostatic stability test was performed on the catalyst

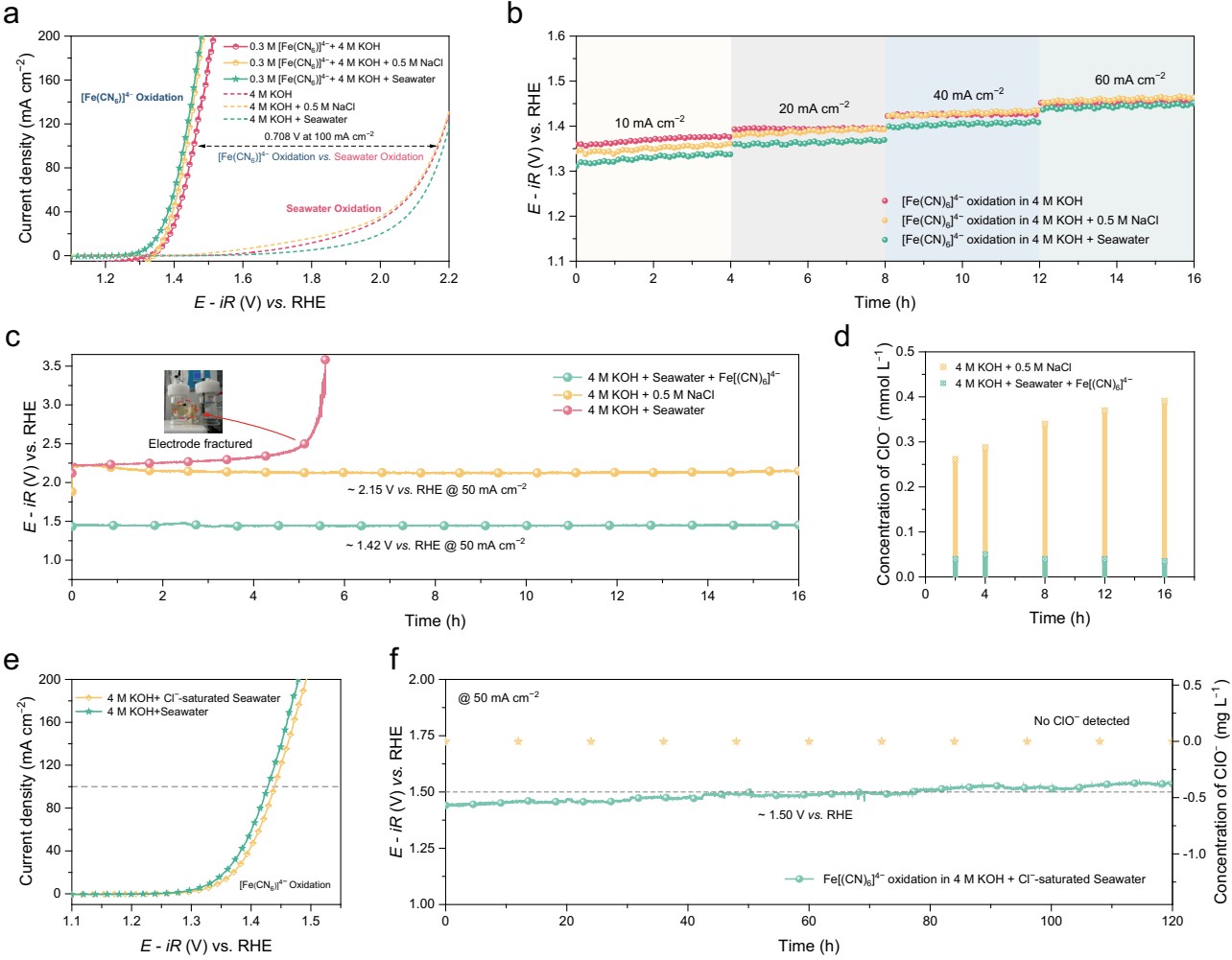

**Fig. 3 | Characterization of the $[Fe(CN)_6]^{4-}$ electrochemical oxidation processes.** The experiments were carried out at room temperature (~25 °C). And the carbon rod electrode acted as the counter electrode, and the Hg/HgO electrode served as the reference electrode. **a** LSV curves of the carbon felt electrode for $[Fe(CN)_6]^{4-}$ oxidation and seawater oxidation with 100% *iR* compensation in different electrolytes at a scan rate of 2 mV s$^{-1}$, where *R* was determined to be 1.2 ± 0.1 Ω. **b** V-T curves for $[Fe(CN)_6]^{4-}$ oxidation in 4 M KOH, 4 M KOH + 0.5 M NaCl, and 4 M KOH + seawater at different current densities. **c** V-T curves of the anodic carbon felt electrode at 50 mA cm$^{-2}$ in three different solutions with/without $[Fe(CN)_6]^{4-}$. **d** Comparison of the ClO$^-$ concentration changes in the anolyte from 0.3 M $[Fe(CN)_6]^{4-}$ + 4 M KOH + seawater and 4 M KOH + 0.5 M NaCl. **e** LSV curves of the carbon felt electrode for $[Fe(CN)_6]^{4-}$ oxidation in 4 M KOH + seawater and 4 M KOH + Cl$^-$-saturated seawater at a scan rate of 2 mV s$^{-1}$ with 100% *iR* compensation, where *R* was determined to be 1.2 ± 0.1 Ω. **f** V-T curve of the carbon felt electrode for $[Fe(CN)_6]^{4-}$ oxidation at 50 mA cm$^{-2}$ in 4 M KOH + Cl$^-$-saturated seawater and accompanying ClO$^-$ concentration changes.

by applying a constant current density of 100 mA cm$^{-2}$ for 100 hours. As shown in Fig. S9c, after 100 h of testing, the catalyst maintained nearly the same overpotential for the HER, confirming its enhanced stability in seawater.

**Verification of the chemical reduction process**

In the redox-mediated decoupled seawater electrolysis system, we utilize the solvent $[Fe(CN)_6]^{4-}$ as an electron donor to suppress the traditional water oxidation reaction in seawater direct electrolysis. During this process, $[Fe(CN)_6]^{4-}$ is converted to $[Fe(CN)_6]^{3-}$ at the anode, as discussed in detail above. Afterward, as the concentration of $[Fe(CN)_6]^{3-}$ increases and the concentration of $[Fe(CN)_6]^{4-}$ decreases, the equilibrium potential of the redox carrier gradually increases (Fig. S10)[30]. Once the potential of the redox carrier reaches the onset potential for the OER, $[Fe(CN)_6]^{3-}$ will undergoes a spontaneous chemical reaction in the presence of a chemical catalyst; it is reduced back to $[Fe(CN)_6]^{4-}$, O$_2$ is generated, and the overall water-splitting reaction is completed. In this process, the optimization and design of the catalyst play pivotal roles in enhancing the chemical reduction reaction of $[Fe(CN)_6]^{3-}$ and OH$^-$; here, a high tolerance in $[Fe(CN)_6]^{3-/4-}$ + alkaline

seawater environments and an efficient oxygen evolution performance of the catalyst are needed.

To achieve this objective, a Fe-doped Ni(OH)$_2$ catalyst was assembled on the surface of nickel foam using a one-step electroplating method, as shown in Fig. 4a (for Fe-Ni(OH)$_2$/NF, see Methods for detailed procedures). SEM images revealed the formation of a mesoporous array of Fe-decorated Ni(OH)$_2$ flower-like particles with average diameters ranging from 800 to 1200 nm (Fig. 4b and Fig. S11). The rough surface of nanoarrays could effectively diminish the gas adhesion at catalytic active sites located at the solid–liquid–gas three-phase interface, consequently achieving a super-hydrophobicity, and ensuring the continuation of the chemical oxygen evolution process[37]. The obtained catalyst was characterized primarily by the α-Ni(OH)$_2$ phase, as evident from the powder X-ray diffraction (XRD) pattern (JCPDS No. 38-0715, Fig. S12). The detailed composition of Fe-Ni(OH)$_2$/NF is shown in the transmission electron microscopy (TEM) image in Fig. S13a; here, the whole flower-shaped nanostructure consists of numerous nanosheets. The high-resolution TEM image (Fig. S13b) further displayed well-resolved lattice fringes with planar spacings of 0.260 and 0.268 nm, which closely fit the (0 1 2) and (1 0 1) lattice planes of α-Ni(OH)$_2$,

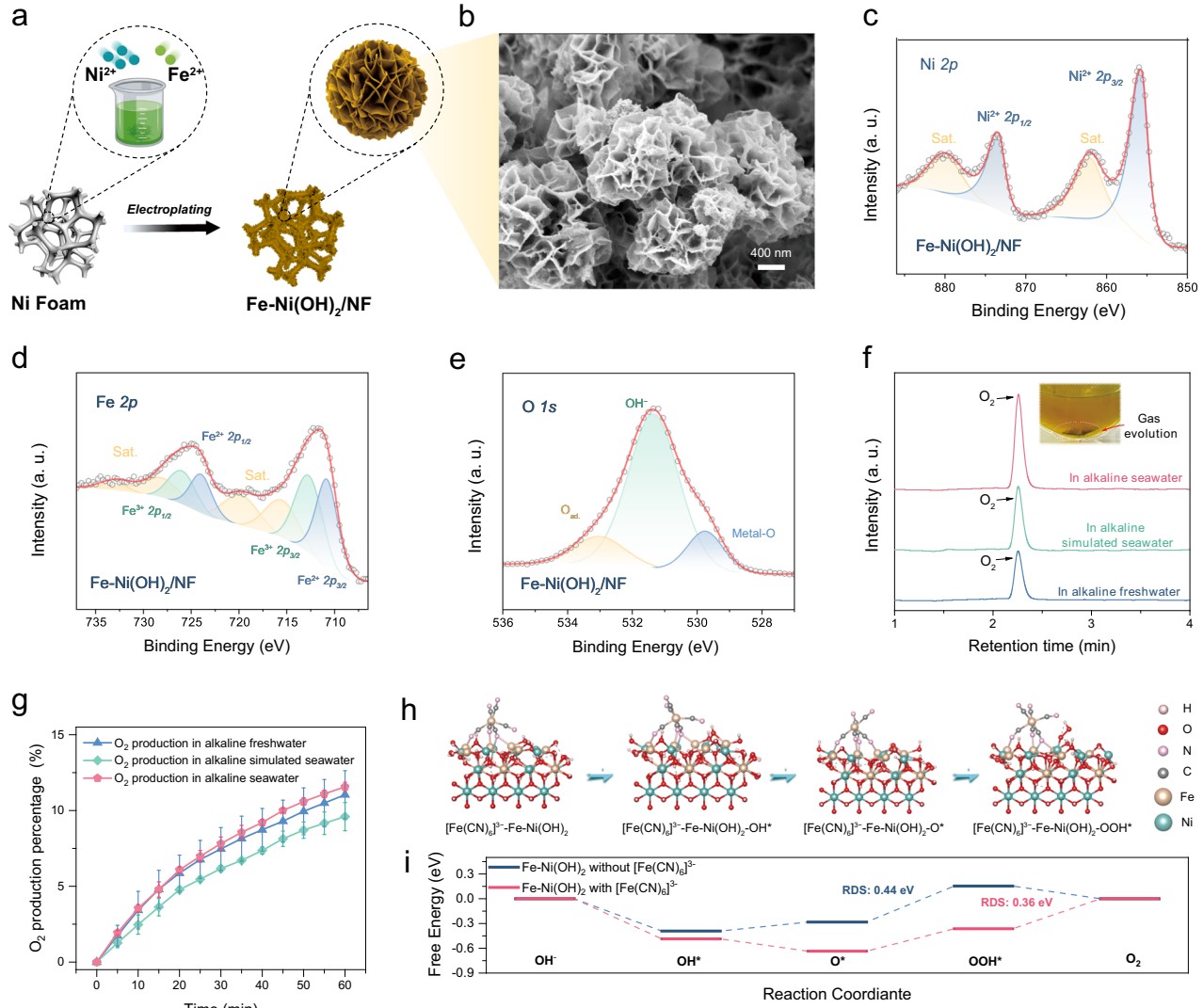

**Fig. 4 | Experimental investigations and theoretical calculations of the chemical reduction reaction. a** Synthetic schematic illustration of Fe-Ni(OH)$_2$/NF. **b** SEM image of Fe-decorated Ni(OH)$_2$ flower-like particles on Ni foam. **c** Ni 2p, **d** Fe 2p, and **e** O 1s XPS spectra of Fe-Ni(OH)$_2$/NF. **f** Gas chromatogram of the produced gases in the three alkaline solutions. **g** Variation in the O$_2$ production over time.

(pi error bar) **h** OER pathways on the Fe-Ni(OH)$_2$/NF catalyst with [Fe(CN)$_6$]$^{3-}$. The silver, red, pink, gray, brown, and green balls represent the H, O, N, C, Fe, and Ni atoms, respectively. **i** Theoretical calculated reaction free energies of the OER process on Fe-Ni(OH)$_2$/NF with/without [Fe(CN)$_6$]$^{3-}$.

respectively. Figure S14 shows a high-angle annular dark-field scanning transmission electron microscopy (HAADF-STEM) image of the catalyst and the corresponding energy dispersive X-ray spectroscopy (EDX) elemental mappings of Ni, Fe, and O, showing their uniform dispersion on the nanosheet of Fe-Ni(OH)$_2$ and verifying the successful doping of Fe. Inductively coupled plasma–optical emission spectroscopy (ICP–OES) analysis provided a more accurate assessment of the Fe doping level in Fe-Ni(OH)$_2$; from these data, the Fe-to-Ni doping ratio was ~19.41:33.31%.

The element composition and electronic configuration of the Fe-Ni(OH)$_2$/NF catalyst were further investigated using X-ray photoelectron spectroscopy (XPS). The XPS survey spectrum (Fig. S15) validated the coexistence of Ni, Fe, and O elements in this catalyst. As illustrated in Fig. 4c, the high-resolution Ni 2p XPS spectrum featured two spin–orbit doublets, and each was accompanied by a corresponding satellite peak, which could be assigned to the characteristic Ni 2p$_{3/2}$ and Ni 2p$_{1/2}$ peaks of Fe-Ni(OH)$_2$/NF. This peak pattern is characteristic of nickel hydroxide (Ni(OH)$_2$)[38]. The Fe 2p XPS spectrum, presented in Fig. 4d, showed peaks at 710.8 and 724.0 eV; these effectively corresponded to the peaks of Fe$^{2+}$ 2p$_{3/2}$ and Fe$^{2+}$ 2p$_{1/2}$, respectively. Additional peaks at

715.5 and 728.1 eV were attributed to their respective satellite peaks. Furthermore, peaks with binding energies of 712.7 and 726.0 eV signified the presence of the Fe$^{3+}$ oxidation state, whereas those at 719.8 and 733.0 eV were associated with the satellite peaks of Fe$^{3+}$ [39]. In the high-resolution O 1s XPS region (Fig. 4e), the binding energies of 529.8, 531.3, and 533.02 eV corresponded to metal−O species, hydroxyl species (OH), and surface-adsorbed water molecules of Ni/Fe−OH; these results confirmed the generation of metal hydroxides[40]. Collectively, these characterizations provided mutual validation for the successful preparation of the Fe-Ni(OH)$_2$/NF catalyst.

After the preparation and characterization of the reduction reaction catalyst, we examined the nature and kinetics of this chemical reduction process in a separate reactor. The gas composition and release efficiency of the reduction solution were assessed; here, 200 mL of 0.15 M [Fe(CN)$_6$]$^{3-}$ and 0.15 M [Fe(CN)$_6$]$^{4-}$ was dissolved in the aforementioned three alkaline solutions at 60 °C and reacted with the Fe-Ni(OH)$_2$/NF catalyst. As shown in Fig. 4f, densely concentrated gas bubbles were observed in the reactor, and only the generation of O$_2$ was identified in the three solutions via gas chromatography. To evaluate the efficiency of gas production, we employed the water

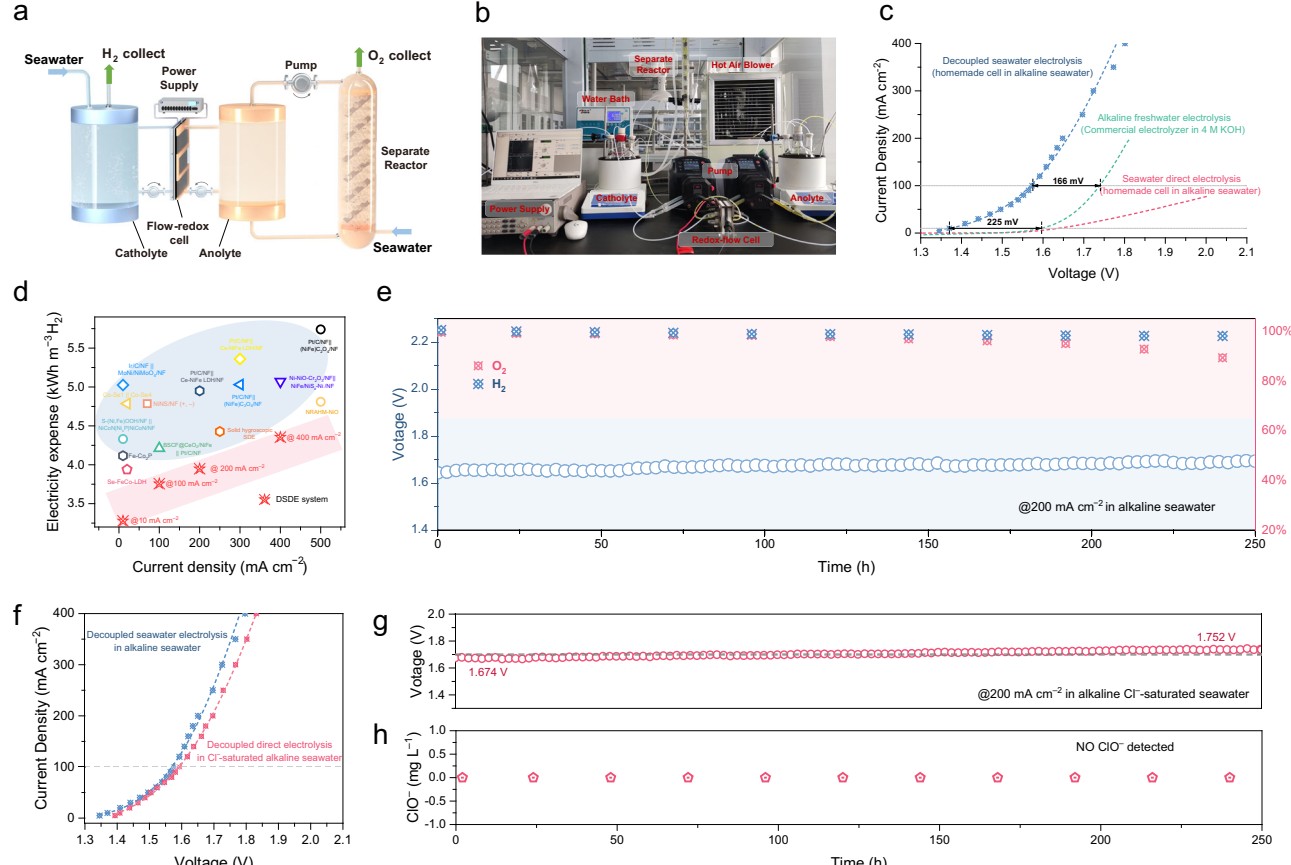

**Fig. 5 | Characterization of the decoupled seawater direct electrolysis system.** **a** Schematic diagram and **b** optical image of the lab-scale DSDE system. **c** LSV curves of the DSDE system, commercial alkaline freshwater electrolyzer, and SDE system. **d** Comparison of the DSDE system with the state-of-the-art seawater electrolyzer at electricity expense vs. current density. **e** Durability test of the DSDE system in real alkaline seawater. **f** LSV curves of the DSDE system in alkaline seawater and Cl⁻-saturated alkaline seawater. **g** Durability test and **h** ClO⁻ concentration change of the DSDE system in Cl⁻-saturated alkaline seawater.

displacement method for one hour. After one hour, the reduction reaction in alkaline seawater yielded 22.93 mL of oxygen; this volume was slightly greater than those released in both the alkaline freshwater and alkaline simulated seawater (Fig. 4g). As depicted in Fig. S16, the production rate gradually decreased in response to the decreasing concentrations of potassium ferricyanide. Based on further experimentation with varying catalyst quantities (Fig. S17a) and temperatures (Fig. S17b), despite the increase in the oxygen evolution rate with increasing catalyst quantity and elevated temperatures, ferricyanide reduction remained the rate-limiting step of the system due to its intrinsic characteristics, such as four-electron transfer and a gas–liquid–solid multiphase thermal catalytic reaction process. However, due to the use of only a single catalyst layer with an area of $1 \times 2$ cm⁻² for catalyzing the reaction, the initial rate of the oxygen release sustained a current of 110 mA for one hour. Hence, attaining a sufficient reduction rate in a continuous flow electrolysis cell becomes feasible, provided that an appropriate amount of catalyst is used for the reduction process and that a stable flow of $[Fe(CN)_6]^{3-/4-}$ is maintained between the reactor and the anode for recharging.

To gain deeper insights into the mechanism of the catalytic reduction reaction, we conducted computational investigations into the reduction process of $[Fe(CN)_6]^{3-}$ and OH⁻ on Fe-Ni(OH)₂/NF to generate O₂ (Original data in Supplementary Data 1). The entire reduction process could be divided into two primary steps, the OER of hydroxide and the electron capture process of $[Fe(CN)_6]^{3-}$. Here, we focused on the O₂ generation process, and computed the OER pathways for the catalyst Fe-Ni(OH)₂/NF with/without $[Fe(CN)_6]^{3-}$. As depicted in Fig. 4h. $[Fe(CN)_6]^{3-}$ initially adhered to the surface of the catalyst, followed by the adsorbate evolution reaction mechanism of the OER, from OH⁻ to OH*, OH* to O*, O* to OOH*, and O₂ generation. In the case of Fe-Ni(OH)₂/NF without $[Fe(CN)_6]^{3-}$ (Fig. S18), the rate-determining step occurred in the transition from O* to OOH* and was characterized by a rate-determining energy barrier of 0.44 eV. However, when the $[Fe(CN)_6]^{3-}$ structure was introduced to the surface of Fe-Ni(OH)₂/NF, we noticed an improvement in the adsorption of intermediates. This modification shifted the rate-determining step to the transition from OOH* to O₂, resulting in a reduced energy barrier of 0.36 eV. Consequently, due to the introduction of $[Fe(CN)_6]^{3-}$ in the reaction, the OER step occurred more easily.

**Spatially decoupled seawater splitting**

Through the systematic evaluation of electrochemical oxidation and chemical reduction processes, we substantiated the performance and efficiency of the two pivotal stages of the decoupled strategy for seawater direct electrolysis. We subsequently designed and implemented a laboratory-scale DSDE system (schematic diagram in Fig. 5a, photograph in Fig. 5b). This system integrated a custom-made redox-flow cell with a separate reactor, and our aim was to provide a comprehensive assessment of the system's overall electrolytic performance. The redox-flow cell had an effective electrochemical area of 4 cm², with a $(NiMo)_{1-x}Co_xP/NF$ electrode on the cathode and a graphite felt electrode at the anode. The catholyte consisted of a 4 M KOH + seawater solution, whereas the anolyte was a blend of 0.15 M $[Fe(CN)_6]^{3-}$ and 0.15 M $[Fe(CN)_6]^{4-}$ dissolved in a 4 M KOH + seawater solution. In the chemical reduction reactor, glass beads and catalysts were placed in the reaction tube to maintain a steady flow, ensuring efficient

solution circulation throughout the system. The LSV curves of the DSDE system, commercial alkaline water electrolyzer and SDE system at 60 °C were measured without *iR* correction (Fig. 5c). At a current density of 10 mA cm$^{-2}$, the DSDE system required only 1.37 V; this value was remarkably 225 mV lower voltage demand than the commercial electrolyzer in alkaline freshwater, which required a voltage of 1.595 V. As the current density increased to 100, 200, and 400 mA cm$^{-2}$, the cell voltages of the DSDE system increased to 1.57, 1.65, and 1.82 V, respectively. This activity significantly surpassed that of the SDE (OER at the anode) on the same custom-made cell. Compared with recent reports on seawater direct electrolysis (Fig. 5d and Table S2), the DSDE system had an advanced level of energy consumption. This advantage was especially evident at low current densities, and the DSDE system required only 3.3 kWh m$^{-3}$ H$_2$ to achieve a current density of 10 mA cm$^{-2}$ since the single-electron kinetics significantly outperformed those of the OER.

During prolonged V-T testing, the DSDE system demonstrated stability, operating continuously for over 250 h at an industrially relevant current density of 200 mA cm$^{-2}$ (Fig. 5e, original chromatograms in Fig. S19 and O$_2$ generation in Supplementary Movie 1). The voltage increased by only 0.05 V throughout the duration, remaining below the onset potential of ClOR. When the Fe-Ni(OH)$_2$ catalyst was observed in the initial stages of the reaction, a rapid transformation from pale yellow to dark brown occurred (Fig. S20). This transition caused the formation of the NiOOH active sites for the OER process[41]. The gas chromatographic tests revealed a declining trend in the O$_2$ evolution Faradaic efficiency after 200 h, reaching ~85% at 240 h. The SEM image of the reduced catalyst after the long-term reaction revealed extensive flushing of the nanoclusters and the appearance of more cracks. A few retained nanoclusters displayed sharp, spike-like morphologies (Fig. S21). The degradation of the catalyst was the main cause of the decrease in the oxygen evolution efficiency. To obtain a more detailed understanding of the degradation from the perspective of material properties, TEM, corresponding EDX, and XPS analyses of Fe-Ni(OH)$_2$ after the long-term reduction reaction was completed. No evident changes were observed in the lattice parameters and element distributions from the TEM, HAADF-STEM, and corresponding EDX images (Figs. S22, S23). The XPS spectra of Fe-Ni(OH)$_2$ after the reaction shown in Fig. S24 reaffirmed the elemental composition and oxidation states but did not show the presence of Ni(III) because of significant nanocluster erosion during the reaction. Thus, to address this issue and to fully exploit the potential within our decoupled system design, the development of efficient chemical reduction catalysts is highly desirable.

In a Cl$^-$-saturated alkaline seawater environment resulting from continuous seawater feeding, the DSDE system exhibits enhanced anti-corrosion properties and maintains stable electrochemical performance. As depicted in Fig. 5f, even at a high Cl$^-$ concentration (3.16 M), the DSDE system exhibited only a modest increase in voltage, reaching 1.59 and 1.70 V at current densities of 100 and 200 mA cm$^{-2}$, respectively. The long-term stability of the DSDE system in Cl$^-$-saturated alkaline seawater is crucial for practical saline water electrolysis. Despite interference from high Cl$^-$ concentrations, the chronopotentiometry results remained stable for more than 250 h, with an average voltage of 1.70 V, and no hypochlorite ions were generated throughout the process (Fig. 5g, h). In addition, no structural corrosion was observed in the carbon felt electrode after long-term electrolysis in Cl$^-$-saturated alkaline seawater (Fig. S25). The Fe-Ni(OH)$_2$ catalyst, after 10 days of V-T testing, was further investigated to explore the impact of Cl$^-$ by using XRD, Raman, and XANES characterization. Given the instability of Ni(III) after oxidation, a reduction procedure in methanol solution was used to revert the catalyst to the Ni(II) valence state. As shown in Fig. S26a, the catalyst maintained the phase of Ni(OH)$_2$ before and after the reaction. And the peak in the Raman spectra didn't shift or disappear evidently (Fig. S26b). Furthermore, in the XANES Ni K-edge spectra (Fig. S26c), the Ni-O bond length showed

no alteration. These characterizations indicate that long-term exposure to a Cl$^-$-saturated alkaline seawater electrolyte doesn't affect the Ni active sites of the catalyst.

In this study, a decoupled seawater direct electrolysis strategy for hydrogen production was introduced. This strategy effectively mitigated the challenging issue of competitive chlorine electro-oxidation reactions. The core concept behind this decoupled strategy involved establishing a redox-mediated electrochemical−chemical cycle to generate oxygen. This cycle led to a more dynamically favorable electro-oxidation reaction against the ClOR on the anode. The [Fe(CN)$_6$]$^{3-/4-}$ couple served as the charge mediator that circulated within this system and had reversible redox kinetics and stability in seawater. Furthermore, an Fe-Ni(OH)$_2$/NF catalyst was synthesized to facilitate the catalytic reduction of [Fe(CN)$_6$]$^{3-}$, resulting in high chemical efficiency. Based on these results, a decoupled seawater direct electrolysis system was developed that incorporated a redox-flow cell integrated with a separate reactor. The system demonstrated consistent operational performance, operating at low voltages, and maintaining stability over extended periods.

Moreover, the design of high-performance redox mediators with high capacity and suitable potential and the development of highly efficient HER electrocatalysts and chemical catalysts for the reduction reaction have the potential to significantly increase the efficiency of decoupled seawater direct electrolysis. These advancements could facilitate a future where seawater could be efficiently and sustainably harnessed as a clean energy source.

## Methods
### Materials
All experimental chemicals used in this work included potassium ferricyanide (K$_3$ [Fe(CN)$_6$], Adamas, 99%), potassium ferrocyanide (K$_4$ [Fe(CN)$_6$], Adamas, 98%), nickel nitrate hexahydrate (Ni(NO$_3$)$_2$·6H$_2$O, Adamas, 99%), ferrous chloride dihydrate (FeCl$_2$·2H$_2$O, Aladdin, 98%), potassium hydroxide (KOH, Aladdin, 95%), acetone (CH$_3$COCH$_3$, Chengdu Colon, 99.5%), and hydrochloric acid (HCl, Chengdu Colon). Nickel foam (Ni Foam, aperture: 110 ppi, area density: 380 ± 20 g cm$^{-2}$), platinum carbon (Pt/C, 20 wt%), anion exchange membranes (FAA-3-PK-75), graphite felt (thickness: ~2 mm), and commercial alkaline water electrolyzers were purchased from SCI Materials Hub. The anion exchange membranes were boiled in 1 M KOH at 80 °C for 12 h and then washed and dried before use. The graphite felts were pretreated by soaking in 98% H$_2$SO$_4$ at 60 °C for 6 h, neutralized, and vacuum-dried for 1 week. O$_2$ (99.999%), Ar (99.999%), and N$_2$ (99.999%) were purchased from the Chengdu Heping Gas Plant. The pumps were purchased from Baoding Lead Fluid Technology Co., Ltd. Deionized water (DI water, resistivity of 18.25 MΩ cm at 25 °C) was used as a solvent and was prepared with a high-purity water machine (ULUP series); additionally, seawater was collected from Shenzhen Bay, China.

### Syntheses of the chemical reduction catalyst Fe-Ni(OH)$_2$/NF
In a typical procedure, 4 mmol of Ni(NO$_3$)$_2$·6H$_2$O and 2 mmol of FeCl$_2$·2H$_2$O were dissolved in 60 mL of DI water to prepare the electrodeposition solution. The working electrode consisted of a piece of Ni foam (1 cm × 2 cm) subjected to ultrasonic cleaning with 4 M HCl, acetone, and deionized water, and the counter electrode was a carbon rod. Subsequently, electrodeposition was conducted at a current density of −10 mA cm$^{-2}$ in an argon environment for 600 s. The as-synthesized NF electrode was rinsed with acetone and DI water and vacuum-dried overnight. The loading weight of the formed electrocatalysts on the nickel foam was ≈15.8 mg cm$^{-2}$.

### Syntheses of the HER electrocatalyst (NiMo)$_{1-x}$Co$_x$P/NF
Ni foam was degreased by sonication in 4 M HCl, acetone, and deionized water. NiMoO$_4$ nanowire alloys on the Ni foam (NiMoO$_4$/NF) were synthesized using a hydrothermal process. 1 mmol NiCl$_2$·6H$_2$O and

1 mmol Na$_2$MoO$_4$·2H$_2$O were dissolved in 15 mL deionized water and stirred vigorously. This solution was then transferred to a Teflon-lined stainless-steel autoclave containing Ni foam and heated to 160 °C for 6 h. After cooling to room temperature, the green solid was washed three times with distilled water and ethanol and then dried in a vacuum oven. Next, 4 mmol 2-methylimidazole was dissolved in 25 mL methanol, and this solution was mixed with methanolic Co(NO$_3$)$_2$·6H$_2$O (1 mmol in 25 mL methanol) while stirring. The NiMoO$_4$/NF was immersed in this mixture and allowed to react for 24 h. After the reaction, the sample was washed with methanol. 20 mg of red phosphorus was placed at the upstream side of a tube furnace, with the (NiMo)$_{1-x}$Co$_x$P/NF precursor placed at the downstream side. The samples were heated to 500 °C at a rate of 2 °C per minute, maintained at this temperature for 90 min in a flow of argon gas, and then cooled to room temperature to produce (NiMo)$_{1-x}$Co$_x$P/NF.

## Characterization

SEM images were captured with a ZEISS Sigma 300 scanning electron microscope. For the HR-TEM images, HAADF-STEM images, and EDX elemental maps, an FEI Talos F200× transmission electron microscope equipped with a Super-X EDX detector was employed. The XRD patterns were recorded using a Bruker D8 Advance diffractometer equipped with a Cu $K_\alpha$ radiation source at a scan angle rate of 1° min$^{-1}$. The catalyst sample was scraped off the Ni foam substrate prior to XRD analysis, and then magnetic separation was employed to remove the residual Ni particles that adhered to the catalyst. XPS analysis was performed using a Thermo Scientific K-Alpha spectrophotometer equipped with a monochromatic Al $K_\alpha$ radiation source. Gaseous products produced during decoupled seawater direct electrolysis were quantified using a Shimadzu GC-2014C in-line gas chromatograph with a thermal conductivity detector (TCD) for O$_2$ and H$_2$ measurement; ultrapure N$_2$ (99.999%) served as the carrier gas. UV–Vis spectroscopy was conducted with a PerkinElmer Lambda 750S ultraviolet spectrophotometer in the wavelength range of 300–600 nm. Raman spectra were carried out by an XploRA PLUS Raman spectrometer equipped with a 532 nm He-Ne laser and a ×50 objective. Ni K-edge analysis was performed with Si(111) crystal monochromators at the BL14W1 beamlines at the Shanghai Synchrotron Radiation Facility (SSRF) (Shanghai, China).

## Electrochemical measurements

**Cyclic voltammetry.** Cyclic voltammetry curves were measured using a three-electrode system at room temperature on a CHI660E electrochemical analyzer. The working electrode was a glassy carbon electrode (diameter: 5 mm, polished before use), with the carbon rod electrode serving as the counter electrode and the Ag/AgCl electrode (in saturated KCl solution) acting as the reference electrode. Ar gas was used to degas the solution for 30 min before and during the tests.

**Rotating disk electrode curves.** Rotating disk electrode curves were obtained using a Metrohm AutoLab PGSTAT302N electrochemical workstation coupled with a Princeton rotating disk electrode. The electrode used was consistent with that employed in the CV curves. The potential range was −0.1 to 0.55 V (vs. Ag/AgCl), with a scan rate of 5 mV s$^{-1}$, and the rotation speed varied from 100 to 2500 r min$^{-1}$. The glassy carbon electrode was placed in a high-purity argon gas atmosphere for 30 min before testing. For each rotation speed, three independent CV experiments were conducted for the derivation of the average value and the determination of the kinetic parameters. The correlation between the limiting current ($i_{lim}$) and the rotation speed ($\omega$) at varying speeds adhered to the Levich equation (Eq. 4), and the diffusion coefficient ($D$) was calculated from the slope of the fitted curve and was $4.496 \times 10^{-6}$ cm$^2$ s$^{-1}$ for [Fe(CN)$_6$]$^{3-/4-}$.

$$i_{lim} = 0.62 \times nFAcD^{\frac{2}{3}}v^{-\frac{1}{6}}\omega^{\frac{1}{2}} \tag{4}$$

Here, the number of transferred electrons ($n$) is 2; the Faraday constant ($F$) is 96,485 C mol$^{-1}$; the effective electrode area ($A$) is 0.196 cm$^2$; the test concentration ($c$) is 10$^{-6}$ mol cm$^{-3}$; the solution dynamic viscosity ($v$) is 0.01 cm$^3$ s$^{-1}$; and $\omega$ represents the rotational angular velocity (rad s$^{-1}$). The relationship between the current value ($i$) and the rotation speed ($\omega$) at different overpotentials follows the Koutecký–Levich equations (Eq. 5).

$$i^{-1} = i_k^{-1} + \left(0.62nFAcD^{\frac{2}{3}}v^{-\frac{1}{6}}\omega^{\frac{1}{2}}\right)^{-1} \tag{5}$$

Equation 6 can be further derived:

$$j = i_0 \times \left(e^{\alpha \times F \times \frac{V}{RT}} - e^{-\beta \times F \times \frac{V}{RT}}\right) \tag{6}$$

Here, ($j$) is the current density, ($i_0$) is the exchange current density, ($\alpha$) and ($\beta$) are the transfer coefficients, ($V$) is the potential difference, ($R$) is the gas constant, and ($T$) is the temperature. The intercept of the curve is ($i_k^{-1}$). The relationship between the limiting diffusion current ($i_k$) and the overpotential ($\eta$) follows the Butler–Volmer equation (Eq. 7).

$$\eta = \frac{2.303RT}{\alpha nF}[log_{10}(i) - log_{10}(i_0)] \tag{7}$$

The other parameters in the equation are consistent with the above description, where ($\alpha$) is the electron transfer coefficient and ($i_0$) is the exchange current and is equal to 0.201 mA cm$^{-2}$ for [Fe(CN)$_6$]$^{3-/4-}$.

**Half-cell electrochemical measurements.** A carbon felt electrode served as the working electrode for the anodic test, a (NiMo)$_{1-x}$Co$_x$P/NF electrode functioned as the working electrode for the cathodic test, and each had a controlled electrode area of 1 cm$^2$. The carbon rod electrode acted as the counter electrode, and the Hg/HgO electrode served as the reference electrode. The Hg/HgO reference electrode was calibrated using a three-electrode system, with Pt foil serving as both the working and counter electrodes. The calibration was performed in an oxygen-free environment within a 0.1 M KOH solution. Electrochemical activation of the reference electrode was achieved through multiple cyclic voltammetry scans until a stable state was reached, followed by zero-point calibration. The LSV curves for the electrochemical oxidation of ferricyanide were recorded in the range of 0.0 to 0.8 V (vs. Hg/HgO) at a slow scan rate of 2 mV s$^{-1}$ to mitigate the capacitive current. All potentials were referenced to the RHE scale by the Nernst equation: E$_{RHE}$ = E$_{Hg/HgO}$ + 0.098 + 0.059 × pH and were adjusted for the $iR$ by E$_{100\%\ iR\ correction}$ = E$_{non\ iR\ correction}$ − $iR$. Electrochemical impedance spectroscopy (EIS) measurements were performed ranging from 0.1 Hz to 100 kHz with an amplitude of 5 mV at open circuit potential. And $R$ is the solution resistance obtained from the EIS results. Triplicate independent experiments were conducted for each electrochemistry test, and the graphical data presented in the manuscript were derived from the mean values of these replicates, with deviations less than 1%. The same three-electrode system was used to measure the redox stability of the oxidation reaction on the carbon felt electrode.

**Decoupled seawater direct electrolysis system measurements.** The DSDE system included a power supply, two peristaltic pumps, a custom-made redox-flow cell, a separate reactor, and two solution tanks for the catholyte and anolyte. The redox-flow cell consists of outer frames, insulation boards, collector channel plates, electrodes, and an anion exchange membrane. An anion exchange membrane (Alkymer, W-75) with a thickness of 75 ± 5 μm was selected. It was soaked in a 1 M NaOH solution at 80 °C for 24 h and then washed with deionized water three to four times before use. The anolyte consisted of 200 mL of 0.15 M [Fe(CN)$_6$]$^{3-}$ and 0.15 M [Fe(CN)$_6$]$^{4-}$ in a 4 M KOH + seawater solution (pH = 14 ± 0.17). The catholyte consisted of 200 mL of 4 M

KOH + seawater solution. The anode electrolyte was circulated through the redox-flow cell and the reactor at a flow rate of 200 mL min$^{-1}$ using a peristaltic pump. Decoupled overall seawater electrolysis was conducted at 60 °C using an ITECH 6911 DC power supply without *iR* compensation. Triplicate independent experiments were conducted for LSV curves, and the data were derived from the mean values of these replicates, with deviations less than 1%.

## Gas measurements

In the $[Fe(CN)_6]^{3-}$ chemical reduction process, the gaseous products were identified by using a gas chromatograph (GC, Shimadzu GC-2014C) equipped with one thermal conductivity detector for the $H_2$ and $O_2$ signals. Then, the products were measured by using a water displacement method. The amount of water in the collection vessel was monitored to determine the volume of oxygen collected, and the volume was recorded at regular 5-min intervals. The actual oxygen production efficiency was obtained by comparing the actual oxygen yield with the theoretical oxygen yield at different temperatures. In a 200 mL solution of 0.15 M $[Fe(CN)_6]^{3-}$, the theoretical maximum oxygen yield is 0.0075 mol. With the assumption of a pressure of 1 bar and constant reaction conditions throughout the experiment, the ideal gas law (Eq. 8) can be used, as follows:

$$pV = nRT \tag{8}$$

where $p$ is the pressure of 1 bar, $V$ is the volume of the gas (L), $n$ is the number of moles of gas, $R$ is the ideal gas constant (0.08314 L bar mol$^{-1}$ K$^{-1}$), and $T$ is the absolute temperature (K). The theoretical maximum oxygen yields at 40, 60, and 80 °C are 189.6, 198.3, and 207.1 mL, respectively.

During the decoupled seawater direct electrolysis, the gaseous products produced were quantified by using the GC results. High-purity nitrogen gas was continuously introduced to collect the oxygen generated in the reduction reactor and the hydrogen produced at the cathode. Chromatographic data were obtained as the average of two replicate experiments. The Faradaic efficiency was obtained by comparing the peak area of the gas chromatogram with the peak area of gas production under standard conditions.

## DFT computation

The density functional theory (DFT) calculations were performed with the VASP package[42]. The zero damping D3 correction method of Grimme[43] was used to describe van der Waals interactions. The generalized gradient approximation (GGA) of Perdew–Burke–Ernzerhof (PBE)[44] was used to calculate the exchange-correlation energies, where the force- and energy-convergence criteria for the self-consistent field (SCF) and energy cutoff were set to $1 \times 10^{-5}$ eV, 0.05 eV Å$^{-1}$, and 450 eV, respectively. A $(3 \times 3 \times 1)$ $k$-point grid was employed for faster convergence during optimization[45]. This slab was separated by a 15 Å vacuum layer in the z direction between the slab and its periodic images. Free energy diagrams for the HER were computed using a computational hydrogen electrode (CHE) model[46]; the results indicated that the chemical potential of a proton/electron ($H^+ + e^-$) was equal to half of that of one $H_2$ gas molecule. The change in free energy ($\Delta G$) for each step in the overall transformation was determined using the following equation (Eq. 9)[46,47]:

$$\Delta G = \Delta E + \Delta ZPE - T\Delta S \tag{9}$$

Where $\Delta E$, $\Delta ZPE$, and $\Delta S$ are the differences in the total energies, zero-point energies, and entropies between the reactant and product, respectively. The zero-point energy values reported by Nørskov et al. were used for all reaction steps, and $\Delta S$ was retrieved from thermodynamic tables at $T = 298$ K.

## Data availability

The data supporting the findings of this study are available in the article and its supplementary files. Any additional requests for information can be directed to, and will be fulfilled by, the corresponding authors. Source data are provided in this paper. Source data are provided with this paper.

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

## Acknowledgements

This work is supported by the Program for the National Natural Science Foundation of China (Grant Nos. 52104400, 52304427, 51827901, 52374133, and U2013603). This work is also supported by the Sustainable Development Technology Special Project of the Shenzhen Science and Technology Innovation Commission (KCXST20221021111601003). We are grateful for the support from the National Key R&D Program of China (Grant No. 2022YFB4102100). We would like to thank Guangdong for Introducing Innovative and Entrepreneurial Teams (Grant No. 2019ZT08G315). We would also like to thank the Sichuan Natural Science Foundation (Grant No. 24NSFSC3635 and No. 24NSFSC6592). We appreciate the Institute of New Energy and Low-Carbon Technology (Sichuan University). We express our gratitude to Professor Xiaopeng Wang for the joint discussions in XAS testing and other tests. In addition, gratitude is extended to Dr. Shijia Mu for the assistance provided during the Raman spectroscopy testing.

## Author contributions

H.X., T.L., and C.L. conceived and designed the idea. T.L., C.L., M.T., M.L., and Y.X. performed the characterizations and experiments. Y.W., C.L., H.Y., Q.D., and M.T. analyzed the data. T.L., C.L., Y.W., and W.J. designed the devices. H.X., T.L., C.L., M.T., W.J., and Z.Z. drafted the article and revised it critically. All the authors reviewed the manuscript.

## Competing interests

The authors declare no competing interests.
