## [Peer Review File · Nature Communications]

Redox-Mediated Decoupled Seawater Direct Splitting for H₂ ProductionREVIEWER COMMENTS

Reviewer #1 (Remarks to the Author):

This paper by Liu et al reports on the development of a decoupled electrolysis system for electrochemical seawater splitting. The idea is to generate hydrogen and oxygen from the process, rather than hydrogen and chlorine-containing species. Decoupled water splitting from seawater is an important challenge and this work produces important strides in making this a reality. The results seem largely supported by the conclusions. Therefore, whilst there are clearly challenges ahead for the work (current densities are still rather low, corrosion and catalyst deactivation issues still require development, etc), I think that there is enough in this work that it merits publication at this juncture. Nature Communications seems to be an appropriate venue for this work. Therefore, my recommendation is to publish once the following comments have been addressed:

1. I do not see any error bars on any graphs, or any error margins quoted on the values for electrochemical or catalytic data that are reported. How many times were the tests repeated? Some mention of number of repeats and associated error margins is definitely required.
2. For the oxygen generation data in Figure S14 and the associated discussion in the main text, it would be good to also express this in terms of a percentage of the total oxygen that would be generated at 100% yield. This will give a better measure of how complete the conversion is than just mL. Is this the slowest step (rate-limiting) on the process? If so, it would be good to mention that. If not, what is the rate-determining step in the overall cycle?
3. There is a mention that oxygen was determined by gas chromatography, but not data is shown. It would be good to show such data, and also data on how hydrogen was detected and quantified. In both cases, gas chromatography is also useful to show that the gases are pure, or if they are not, to show what the impurities are and their amounts.

Reviewer #2 (Remarks to the Author):

The authors have introduced a decoupled seawater direct electrolysis (DSDE) strategy where a ferricyanide/ferrocyanide ($[\text{Fe}(\text{CN})_6]^{3-/4-}$) redox mediator is integrated with a separate O_2 evolution reactor, and it effectively suppresses the ClOR. This DSDE cell requires a low cell voltage of 1.57 V at 100 mA cm^{-2} , maintaining stability even in Cl-saturated alkaline seawater. The catalyst is well known [10.1021/acseenergylett.9b00047; 10.1007/s12274-020-3164-3]. However, the work is decent and shows a way to generate H_2 directly from alkaline seawater without ClOR. I would recommend considering this work for publication, after major revisions are made based on the following comments.

1) The reactions have been performed in Cl-saturated alkaline seawater, and the authors have claimed a remarkable anti-corrosion ability. What is impact of chloride ions on the Fe-Ni(OH) $_2$ catalyst after prolonged exposure to this electrolyte media? To validate the claim, operando DRIFT and/or XANES/EXAFS data are needed to unravel the changes in bonding characteristics and electronic environment of the metal sites.

2) The operando spectral data should also be supported by a computational understanding.

3) The authors synthesized Fe doped Ni(OH) $_2$ on Ni foam. However, the provided characterization does not support the doping of Fe into Ni(OH) $_2$. What is the percentage of Fe into the Ni(OH) $_2$? EDXS elemental mapping in Figure S12, suggests that the density of Fe is more than Ni.

4) In Figure 5e,g,h, the durability tests should be extended up to a minimum of 10 days, to show the viability of this process.

5) In Figure 3c, while performing long term stability by chronopotentiometry at 50 mA cm^{-2} , the corresponding voltage is ~ 2.15 V vs. RHE in 4M KOH + 0.5 M NaCl, and ~ 2.5 V in 4M KOH + seawater. However, the LSV plot in Figure 3a, suggests that it should be ~ 1.7 V vs. RHE.

6) Chronopotentiometry in alkaline simulated seawater is stable for 16 hours, whereas in alkaline seawater, it is less than 1 hour (Figure 3c). Why is there such a difference in performance although electrochemical corrosion by Cl- is present in both the cases.

7) In Figure S7a, the electrocatalytic HER performance in alkaline fresh water, and alkaline sea water shows the generation of ~ -6 , and -3 mA cm^{-2} current density at 0 V vs. RHE, respectively. Appearance of this significant current at 0 V cannot be from HER. The authors should address this.

8) Why is there no peak for the Ni foam substrate in the XRD pattern of Fe-Ni(OH)₂/NF (Figure S10)?

9) The authors should rectify the anode/cathode labeling in Figure 1a.

10) There are quite a number of typos in this manuscript.

Point-by-point responses to the reviewers' comments

Title: Redox-Mediated Decoupled Seawater Direct Splitting for H₂ Production

Manuscript ID: NCOMMS-24-12522-T

Referee #1

Comments :

This paper by Liu et al reports on the development of a decoupled electrolysis system for electrochemical seawater splitting. The idea is to generate hydrogen and oxygen from the process, rather than hydrogen and chlorine-containing species. Decoupled water splitting from seawater is an important challenge and this work produces important strides in making this a reality. The results seem largely supported by the conclusions. Therefore, whilst there are clearly challenges ahead for the work (current densities are still rather low, corrosion and catalyst deactivation issues still require development, etc.), I think that there is enough in this work that it merits publication at this juncture. Nature Communications seems to be an appropriate venue for this work. Therefore, my recommendation is to publish once the following comments have been addressed.

Response:

We would like to express our sincere gratitude for your constructive feedback and the recognition of the importance and potential impact of this work. We appreciate your positive evaluation of our findings and the identification of the challenges that lie ahead. We have reviewed and addressed your comments as follows.

Comments :

1. I do not see any error bars on any graphs, or any error margins quoted on the values for electrochemical or catalytic data that are reported. How many times were the tests repeated? Some mention of number of repeats and associated error margins is definitely required.

Response:

Thank you for your valuable comments regarding the importance of data repeatability and error margins in our study. Your comments are crucial for enhancing the rigor and credibility of our research. In response to your concerns, we have made the following additions and revisions:

1. For the majority of the electrochemical assays, triplicate independent experiments were executed to ensure the precision of our findings. The graphical data presented in the manuscript are derived from the mean values of these replicates. In this revision, we have included the following additional

sets of three independent replicate experiments: the seawater oxidation LSV curves and I-T stability, as well as the HER LSV curves for the $(\text{NiMo})_{1-x}\text{Co}_x\text{P/NF}$ catalyst, to avoid errors caused by experimental variability as much as possible.

2. We have included error bars in some graphs to represent the variability of the data such as those in Fig. 4g, Fig. 5c and Fig. 5f, in the revised manuscript. In this amended submission, we have added error bars to the newly added experiments: ferricyanide oxidation reactions with different catalyst amounts (Fig. R1, Fig. S15a in the revised manuscript) and at different temperatures (Fig. R2, Fig. S15b in the revised manuscript), which are described in detail in the following commentary.

3. Furthermore, we have also added descriptions to the “Methods” section regarding the repeatability of the experiments (lines 25-27, page 18).

‘Triplicate independent experiments were conducted for each electrochemistry test, and the graphical data presented in the manuscript were derived from the mean values of these replicates.’

4. To improve the transparency and traceability of our research, we will include all raw datasets in the final uploaded version as supplementary material for peer review and reader reference.

Comments :

2. *For the oxygen generation data in Figure S14 and the associated discussion in the main text, it would be good to also express this in terms of a percentage of the total oxygen that would be generated at 100% yield. This will give a better measure of how complete the conversion is than just mL. Is this the slowest step (rate-limiting) on the process? If so, it would be good to mention that. If not, what is the rate-determining step in the overall cycle?*

Response:

Thank you for your meticulous comments and suggestions, which have been instrumental in enhancing the clarity and precision of our data presentation. In response to your feedback, we have revised the manuscript (Fig. R1 and Fig. R2, Fig. 4g and Fig. S14 in the revised manuscript) to express the oxygen generation data as a percentage of the theoretical yield at 100% conversion and to provide a standardized measure of reaction completeness.

Figure R1. Variation in the O₂ production over time.
(*Figure 4(g) in the revised manuscript*)

Figure R2. O₂ production rates over time in different solutions.
(*Figure S14 in the revised manuscript*)

Regarding your inquiry on the rate-limiting step, we concur with your perspective that the ferricyanide reduction reaction for oxygen evolution is the rate-determining step in the DSDE cycle. The DSDE system consists of three processes: HER at the cathode, ferrocyanide electrochemical oxidation at the anode, and ferricyanide reduction in a separate reduction reactor. For the electrocatalytic reactions, from the LSV curves of the two reactions (Fig. 3a and Fig. S7a in the revised manuscript), the rapid current increases with increasing voltage; these results indicate that the reaction rate can be quickly increased. In contrast, ferricyanide reduction for oxygen evolution is a four-electron transfer, gas–liquid–solid multiphase thermal catalytic redox reaction, which is quite different. In electrochemical reactions, electrons transfer occurs at distinct locations, whereas in chemical redox reactions, electron transfer occurs at the same site. Consequently, electron exchange in the ferricyanide reduction reaction is more complex, which is the reason for the inherently limited rate for the ferricyanide reduction reaction.

Furthermore, to elucidate this deduction further, additional experiments were conducted. We added

and compared the amount of oxygen produced in seawater containing potassium ferrocyanide by varying both the catalyst quantity (Fig. R3a, Fig. S15a in the revised manuscript) and the temperature (Fig. R3b, Fig. S15b in the revised manuscript). While an increase in the catalyst amount and temperature moderately increased the oxygen evolution rate, the improvement was markedly less than that of the electrocatalytic reactions, highlighting ferricyanide reduction as the overarching rate-limiting step.

Figure R3. Variation in the O₂ production over time using (a) different amounts of the catalyst and (b) different temperatures.

(Figure S15 in the revised manuscript)

Consequently, we have augmented the manuscript at lines 16-21, page 11 with the following explanation:

'Based on further experimentation with varying catalyst quantities (Fig. S15a) and temperatures (Fig. S15b), despite the increase in the oxygen evolution rate with increasing catalyst quantity and elevated temperatures, ferricyanide reduction remained the rate-limiting step of the system due to its intrinsic characteristics, such as a four-electron transfer and a gas-liquid-solid multiphase thermal catalytic reaction process.'

Comments :

3. There is a mention that oxygen was determined by gas chromatography, but not data is shown. It would be good to show such data, and also data on how hydrogen was detected and quantified. In both cases, gas chromatography is also useful to show that the gases are pure, or if they are not, to show what the impurities are and their amounts.

Response:

Thank you for your constructive comments and valuable suggestions. Following your suggestion,

we have incorporated the original chromatograms for both oxygen and hydrogen evolution during decoupled seawater direct electrolysis into the revised manuscript (Fig. R4, Fig. S17 in the revised manuscript).

Figure R4. Original chromatograms for (a) hydrogen and (b) oxygen evolution during decoupled seawater direct electrolysis.

(Figure S17 in the revised manuscript)

Additionally, we have supplemented the “Methods” section (in lines 4-24, page 19) with detailed descriptions of the gas evolution measurement by the water displacement method and gas chromatograph analysis:

‘Gas measurement

In the $[\text{Fe}(\text{CN})_6]^{3-}$ chemical reduction process, the gaseous products were identified by using a gas chromatograph (GC, Shimadzu GC-2014C) equipped with one thermal conductivity detector for the H₂ and O₂ signals. Then, the products were measured by using a water displacement method. The amount of water in the collection vessel was monitored to determine the volume of oxygen collected, and the volume was recorded at regular 5-min intervals. The actual oxygen production efficiency was obtained by comparing the actual oxygen yield with the theoretical oxygen yield at different temperatures. In a 200 mL solution of 0.15 M $[\text{Fe}(\text{CN})_6]^{3-}$, the theoretical maximum oxygen yield

is 0.0075 mol. With the assumption of a pressure of 1 bar and constant reaction conditions throughout the experiment, the ideal gas law can be used, as follows:

$$pV = nRT$$

where p is the pressure of 1 bar, V is the volume of the gas (L), n is the number of moles of gas, R is the ideal gas constant ($0.0821 \text{ L atm mol}^{-1} \text{ K}^{-1}$), and T is the absolute temperature (K). The theoretical maximum oxygen yields at 40°C , 60°C , and 80°C are 189.6, 198.3, and 207.1 mL, respectively.

During the decoupled seawater direct electrolysis, the gaseous products produced were quantified by using the GC results. High-purity nitrogen gas was continuously introduced to collect the oxygen generated in the reduction reactor and the hydrogen produced at the cathode. Chromatographic data were obtained as the average of two replicate experiments. The Faradaic efficiency was obtained by comparing the peak area of the gas chromatogram with the peak area of gas production under standard conditions.'

Referee #2

Comments :

The authors have introduced a decoupled seawater direct electrolysis (DSDE) strategy where a ferricyanide/ferrocyanide ($[\text{Fe}(\text{CN})_6]^{3-/4-}$) redox mediator is integrated with a separate O_2 evolution reactor, and it effectively suppresses the ClOR. This DSDE cell requires a low cell voltage of 1.57 V at 100 mA cm^{-2} , maintaining stability even in Cl^- -saturated alkaline seawater. The catalyst is well known [10.1021/acseenergylett.9b00047; 10.1007/s12274-020-3164-3]. However, the work is decent and shows a way to generate H_2 directly from alkaline seawater without ClOR. I would recommend considering this work for publication, after major revisions are made based on the following comments.

Response:

We are truly grateful to you for your positive evaluation and insightful feedback on our decoupled seawater direct electrolysis (DSDE) strategy. We have enhanced the quality and strength of our work based on your valuable comments to ensure that the manuscript meets the high standards expected of *Nature Communications*.

Comments :

- 1. The reactions have been performed in Cl^- -saturated alkaline seawater, and the authors have claimed a remarkable anti-corrosion ability. What is impact of chloride ions on the $\text{Fe-Ni}(\text{OH})_2$ catalyst after prolonged exposure to this electrolyte media? To validate the claim, operando DRIFT and/or XANES/EXAFS data are needed to unravel the changes in bonding characteristics and electronic environment of the metal sites.*
- 2. The operando spectral data should also be supported by a computational understanding.*

Response:

We greatly appreciate your insightful comments and suggestions. The impact of chloride ions on $\text{Fe-Ni}(\text{OH})_2$ catalyst in long-term Cl^- -saturated alkaline seawater is an important issue that requires further investigation via supplementary characterization.

The $\text{Fe-Ni}(\text{OH})_2$ catalyst after undergoing 10-days V-T testing was further investigated. XRD, Raman, and XANES characterizations were conducted to explore any changes in the bonding characteristics and electronic environment of the Ni sites in the $\text{Fe-Ni}(\text{OH})_2$ before and after prolonged reaction. Given the instability of Ni(III) after oxidation, we followed the reduction

procedure in methanol solution, as referenced in *Energy Environ. Sci.*, 2023, 16, 641 and *Energy Environ. Sci.*, 2020, 13, 229, to revert the catalyst to the Ni(II) valence state for the aforementioned characterization. As shown in Fig. R5 (Fig. S24a in the revised manuscript), the catalyst maintained the phase of Ni(OH)₂ before and after the reaction. And the peak in the Raman spectra didn't shift or disappear evidently (Fig. R6, Fig. S24b in the revised manuscript). Furthermore, in the XANES Ni K-edge spectra (Fig. R7, Fig. S24c in the revised manuscript), the Ni-O bond length showed no alteration. These characterizations indicate that long-term exposure to a Cl⁻-saturated alkaline seawater electrolyte doesn't affect the Ni active sites of the catalyst.

Additionally, as you mentioned, Fe-Ni(OH)₂ is a commonly used catalyst. We have further corroborated our findings by reviewing the literature, such as *Advanced Functional Materials*, 2022, 32(29): 2200951; *Energy & Environmental Science*, 2020, 13(6), 1725-1729; *Nature Sustainability*, 2024, 7, 158-167. Nickel-iron hydroxide-based catalysts have demonstrated excellent stability in seawater environments, with anionic intercalation that effectively mitigates interference with the catalyst.

Figure R5. XRD pattern of Fe-Ni(OH)₂/NF before and after reaction.
(*Figure S24a in the revised manuscript*)

Figure R6. Raman spectra of Fe-Ni(OH)₂/NF before and after reaction.
(*Figure S24b in the revised manuscript*)

Figure R7. Ni K-edge XANES spectra of Fe-Ni(OH)₂/NF before and after reaction.
(*Figure S24c in the revised manuscript*)

Therefore, we have augmented the manuscript at lines 17-25, page 14 with the following explanation:

'The Fe-Ni(OH)₂ catalyst after 10-days V-T testing was further investigated to explore the impact of Cl⁻ by using XRD, Raman, and XANES characterizations. Given the instability of Ni(III) after oxidation, a reduction procedure in methanol solution was used to revert the catalyst to the Ni(II) valence state. As shown in Fig. S24a, the catalyst maintained the phase of Ni(OH)₂ before and after the reaction. And the peak in the Raman spectra didn't shift or disappear evidently (Fig. S24b). Furthermore, in the XANES Ni K-edge spectra (Fig. S24c), the Ni-O bond length showed no alteration. These characterizations indicate that long-term exposure to a Cl⁻-saturated alkaline seawater electrolyte doesn't affect the Ni active sites of the catalyst.'

Comments :

3. *The authors synthesized Fe doped Ni(OH)₂ on Ni foam. However, the provided characterization does not support the doping of Fe into Ni(OH)₂. What is the percentage of Fe into the Ni(OH)₂? EDXS elemental mapping in Figure S12, suggests that the density of Fe is more than Ni.*

Response:

Thank you for your detailed review. The actual composition of the catalyst is crucial. To address your concerns, the molar ratio of iron to nickel in the synthesis was set to 1: 2. While the EDX maps in Figure S12 indicate areas where the density of Fe appears higher, this does not necessarily reflect the overall doping percentage of the entire catalyst sample. EDX analysis is influenced by various factors, including the sample's geometry, thickness, and elemental composition, and is typically used to determine the elemental composition and ratio on the surface of a catalyst.

To obtain a more accurate assessment of the Fe doping level in Fe-Ni(OH)₂, we conducted ICP–OES analysis using two methods. In the first method, we scraped the catalyst layer off the Ni foam substrate and used a magnet to remove nickel particles from the catalyst to obtain the test sample. In the second method, we directly electrodeposited the Fe-Ni(OH)₂ catalyst onto a carbon cloth for testing. The elemental analysis results of the two methods indicated a doping ratio of Fe to Ni of approximately 19.41%: 33.31% (from the scraped sample) and 1.88%: 3.21% (from carbon cloth sample). We have now included these detailed findings in the revised manuscript to provide a clear and precise representation of the doping levels (in lines 22-24, page 10):

‘Inductively coupled plasma–optical emission spectroscopy (ICP–OES) analysis provided a more accurate assessment of the Fe doping level in Fe-Ni(OH)₂; from these data, the Fe-to-Ni doping ratio was approximately 19.41%:33.31%.’

Comments :

4. *In Figure 5e, g, h, the durability tests should be extended up to a minimum of 10 days, to show the viability of this process.*

Response:

We are very grateful to you for your suggestions, as they are crucial for improving the strength and quality of our paper. To demonstrate the viability of this process, we have optimized the AEM and reconstructed a decoupled seawater direct electrolysis (DSDE) system (Fig. R8b, Fig. 5b in the revised manuscript). The entire system was maintained at approximately 60°C using a hot air blower and a hot water bath. Under both alkaline seawater and Cl⁻-saturated alkaline seawater conditions, our DSDE system operated for more than 250 hours (Fig. R8e and Fig. R8g, Fig. 5e and Fig. 5g in the revised manuscript), with an overall hydrogen production efficiency of 95% and a final oxygen production efficiency of 85% without the generation of ClO⁻. The SEM analysis of the carbon felt electrode and Fe-Ni(OH)₂/NF catalyst after the extended durability test revealed no significant corrosion or structural changes in the carbon felt fibers (Fig. R9, Figs. S23 in the revised manuscript). Based on these experimental results, the DSDE system exhibits durability exceeding 250 hours and has the potential for long-term stable hydrogen production from seawater electrolysis. Therefore, we have also revised the description of this part: “Spatially decoupled seawater splitting” in the manuscript.

Figure R8 Characterization of the decoupled seawater direct electrolysis system. (a) Schematic diagram and (b) an optical image of the lab-scale DSDE system. (c) LSV curves of the DSDE system, commercial alkaline freshwater electrolyzer and SDE system. (d) Comparison of the DSDE system with the state-of-the-art seawater electrolyzer at electricity expense *vs.* current density. (e) Durability test of the DSDE system in real alkaline seawater. (f) LSV curves of the DSDE system in alkaline seawater and Cl⁻-saturated alkaline seawater. (g) Durability test and (h) ClO⁻ concentration change of the DSDE system in Cl⁻-saturated alkaline seawater.

(*Figure 5 in the revised manuscript*)

Figure R9 (a) High-resolution and (b) low-resolution SEM images of the carbon felt electrode before the long-term electrolysis in Cl⁻-saturated alkaline seawater. (c) High-resolution and (d) low-resolution SEM images of the carbon felt electrode after long-term electrolysis in Cl⁻-saturated alkaline seawater.

(*Figure S23 in the revised manuscript*)

Comments :

5. In Figure 3c, while performing long term stability by chronopotentiometry at 50 mA cm^{-2} , the corresponding voltage is $\sim 2.15 \text{ V}$ vs. RHE in $4 \text{ M KOH} + 0.5 \text{ M NaCl}$, and $\sim 2.5 \text{ V}$ in $4 \text{ M KOH} + \text{seawater}$. However, the LSV plot in Figure 3a, suggests that it should be $\sim 1.7 \text{ V}$ vs. RHE.

Response:

We greatly appreciate your professional review and for pointing out the error in the electrochemical oxidation reaction of the carbon felt electrode. We realized that when obtaining the OER-LSV curves in Fig. 3a, we neglected the oxidation of the carbon felt itself in high-concentration alkaline electrolytes. This oversight led to tests of the electrode without sufficient activation, resulting in data that did not accurately reflect the direct oxidation reaction of the carbon felt electrode.

Now, we ensured full activation of the carbon felt electrode and remeasured LSV curves for electro-oxidation in three electrolytes. As shown in Fig. R10 (Fig. 3a in the revised manuscript), at a current density of 50 mA cm^{-2} , the potential of the carbon felt electrode in $4 \text{ M KOH} + 0.5 \text{ M NaCl}$ was 2.10 V , and in $4 \text{ M KOH} + \text{seawater}$ was 2.11 V ; this value closely aligned with the results shown in Fig. R11 (Fig. 3c in the revised manuscript).

Additionally, to verify the accuracy of the long-term stability experiments, we reconducted three sets of V-T tests at a current density of 50 mA cm^{-2} for the carbon felt electrode in $4 \text{ M KOH} + 0.5 \text{ M NaCl}$ and $4 \text{ M KOH} + \text{seawater}$. In $4 \text{ M KOH} + 0.5 \text{ M NaCl}$, all three electrodes were subjected to direct electrolysis for more than 16 hours. However, in $4 \text{ M KOH} + \text{seawater}$, all three electrodes eventually fractured, leading to a rapid increase in the reaction potential. We used the median fracture time from these tests to plot the relevant curves in Fig. R11.

We have revised the data in Fig. 3a and Fig. 3c in the revised manuscript accordingly and provided updated charts and detailed experimental results in the revised manuscript.

Figure R10 LSV curves of the carbon felt electrode for $[\text{Fe}(\text{CN})_6]^{4-}$ oxidation and seawater oxidation with different electrolytes. (Figure 3 (a) in the revised manuscript)

Figure R11 V-T curves of the anodic carbon felt electrode at 50 mA cm^{-2} in three different solutions with/without $[\text{Fe}(\text{CN})_6]^{4-}$.
(*Figure 3 (c) in the revised manuscript*)

Comments :

- Chronopotentiometry in alkaline simulated seawater is stable for 16 hours, whereas in alkaline seawater, it is less than 1 hour (Figure 3c). Why is there such a difference in performance although electrochemical corrosion by Cl^- is present in both the cases.

Response :

Thank you for your insightful comments. Despite the presence of Cl^- electrochemical corrosion in both cases, the pronounced difference in the stability performance between alkaline simulated seawater and alkaline seawater sample is attributed primarily to the accelerated corrosion and subsequent fracture of the carbon felt electrode in alkaline seawater.

To verify this, we conducted three independent experiments, each resulting in fracture of the carbon felt electrode, which was followed by a rapid increase in the reaction potential. We also provided photographic evidence of the carbon felt electrode at various times during electrolysis (Fig. R12, Fig. S4 in the revised manuscript). These images illustrate the transition from an initially compact state to a loose and curved configuration, culminating in fracture.

Figure R12 Photos of the carbon felt electrode at various times during electrolysis.
(*Figure S4 in the revised manuscript*)

Moreover, in the natural seawater environment, a multitude of complex ions, organic matter, and microorganisms are present (the composition of the seawater used in this study is detailed in Table S1). Compared with those in simulated alkaline seawater, the combined corrosive effects of these complex components in seawater are likely contributed to the increased possibility of the catalyst fracturing. For example, bromide ions (Br^-), as identified by Zhang et al., can lead to extensive corrosion, forming broad and shallow pits, whereas Cl^- results in localized pitting corrosion with narrow and deep pits (*Nature Communications*, 2023, 14(1): 4822). Additionally, trace residues of Ca^{2+} and Mg^{2+} remaining after the alkalization process may precipitate on the catalyst surface during electrolysis, further contributing to the observed electrode fracture.

Comments :

7. *In Figure S7a, the electrocatalytic HER performance in alkaline fresh water, and alkaline sea water shows the generation of ~ -6 , and -3 mA cm^{-2} current density at 0 V vs. RHE, respectively. Appearance of this significant current at 0 V cannot be from HER. The authors should address this.*

Response:

Thank you for your careful review and for pointing out the unexpected current densities observed at 0 V vs. RHE in the electrocatalytic HER performance. We concur with your statement that the significant current densities appearing at 0 V should not be attributed to the HER process. Due to its high porosity and large specific surface area, the Ni foam-substrate electrode facilitates a notable double-layer capacitance that contributes to the formation of capacitive currents, which accounts for the current densities observed in our study.

Although we initially employed a slow scan rate of 2 mV s^{-1} to minimize the capacitive current, the results were not satisfactory. To address this issue, we conducted tests under quasi-static conditions by obtaining LSV curves with an even slower scan rate of 0.1 mV s^{-1} . We have revised Figure R13 (Figure S7 (a) in the revised manuscript) with the retested data, and this approach has yielded significantly improved results, effectively reducing the impact of the capacitive currents and more accurately presenting the HER activity.

Figure R13 Polarization curves of $(\text{NiMo})_{1-x}\text{Co}_x\text{P/NF}$ for the HER in alkaline freshwater and alkaline seawater.

(Figure S7 (a) in the revised manuscript)

Comments :

8. *Why is there no peak for the Ni foam substrate in the XRD pattern of Fe-Ni(OH)₂/NF (Figure S10)?*

Response:

Thank you for your insightful observation regarding the absence of a peak for the Ni foam substrate in the XRD pattern of Fe-Ni(OH)₂/NF presented in Figure S10. In the XRD results, the presence of the Ni foam substrate can lead to a prominent Ni peak, which overshadows the characteristic peaks of other substances and causes difficulty to detect other peaks with lower intensities. To address this interference, we carefully scraped the catalyst layer off the Ni foam substrate prior to XRD analysis. We then employed magnetic separation to remove the residual Ni particles that adhered to the catalyst. This process ensured that the XRD pattern reflected the phase composition of the Fe-Ni(OH)₂ material without being affected by the substrate. We have now included a detailed description of our sample preparation procedure in the manuscript to ensure clarity and reproducibility in lines 6-8, page 17.

“The catalyst sample was scraped off the Ni foam substrate prior to XRD analysis and then magnetic separation was employed to remove the residual Ni particles that adhered to the catalyst.”

Comments :

9. *The authors should rectify the anode/cathode labeling in Figure 1a.*

Response:

Thank you for your careful review and for pointing out the discrepancy in the labeling of the anode and cathode in Fig. 1a. We apologize for this oversight and understand the importance of accurate labeling in scientific figures. We have promptly rectified the labeling in Figure 1a in the revised version and have also reviewed the remaining manuscript to ensure that there are no other labeling inconsistencies or errors.

Comments :

10. There are quite a number of typos in this manuscript.

Response:

Thank you for your thorough review of our manuscript and your attention to detail, which have helped us improve the manuscript. We understand that such typographical errors can detract from the overall quality and readability of a scientific paper. Therefore, we carefully proofread the entire manuscript, corrected typos, and scrutinized the manuscript to improve the language through the use of the language polishing service (Fig. R14).

Figure R14 Editing Certificate of the manuscript.

REVIEWERS' COMMENTS

Reviewer #2 (Remarks to the Author):

The revisions are satisfactory. I recommend the revised manuscript for acceptance in the journal.